# Recent Advances in the Specialized Metabolites Mediating Resistance to Insect Pests and Pathogens in Tea Plants (*Camellia sinensis*)

**DOI:** 10.3390/plants13020323

**Published:** 2024-01-22

**Authors:** Jin Zhang, Yongchen Yu, Xiaona Qian, Xin Zhang, Xiwang Li, Xiaoling Sun

**Affiliations:** Key Laboratory of Tea Biology and Resources Utilization, Ministry of Agriculture, Tea Research Institute, Chinese Academy of Agricultural Sciences, Hangzhou 310008, China; zhangjin1369@tricaas.com (J.Z.); w935369897@163.com (Y.Y.); qxnqxn0412@163.com (X.Q.); xinzhang@tricaas.com (X.Z.); lixiwang0392@tricaas.com (X.L.)

**Keywords:** *Camellia sinensis*, tea plant, specialized metabolites, defense response, insect pest attack, pathogen infection

## Abstract

Tea is the second most popular nonalcoholic beverage consumed in the world, made from the buds and young leaves of the tea plants (*Camellia sinensis*). Tea trees, perennial evergreen plants, contain abundant specialized metabolites and suffer from severe herbivore and pathogen attacks in nature. Thus, there has been considerable attention focusing on investigating the precise function of specialized metabolites in plant resistance against pests and diseases. In this review, firstly, the responses of specialized metabolites (including phytohormones, volatile compounds, flavonoids, caffeine, and L-theanine) to different attacks by pests and pathogens were compared. Secondly, research progress on the defensive functions and action modes of specialized metabolites, along with the intrinsic molecular mechanisms in tea plants, was summarized. Finally, the critical questions about specialized metabolites were proposed for better future research on phytohormone-dependent biosynthesis, the characteristics of defense responses to different stresses, and molecular mechanisms. This review provides an update on the biological functions of specialized metabolites of tea plants in defense against two pests and two pathogens.

## 1. Introduction

Due to their sessile nature, plants have an innate immune system that helps them defend against different pathogen infections and insect attacks, and the defense response is composed of a highly regulated and complex molecular network [1]. After infection by pathogens, plants can initiate two branches of immunity, including pathogen-associated molecular pattern (PAMP)-triggered immunity (PTI) and effector-triggered immunity (ETI) [2]. In the process of PTI and ETI, on the one hand, after a local infection by pathogens, plants can produce a long-lasting resistance to broad-spectrum pathogens in uninfected distal tissues, and this type of resistance in the whole plant is termed systemic acquired resistance (SAR) [3], and on the other hand, beneficial microorganisms are employed to induce plant resistance by hormone signaling or modulating host small RNAs, termed induced systemic resistance (ISR) [4]. For example, the utilization of plant growth-promoting rhizobacteria (PGPR) strains to activate ISR against two major root diseases in tea plants [5]. Upon insect attack, plants respond to both herbivore-associated molecular patterns (HAMPs) and plant-derived damage-associated molecular patterns (DAMPs) and trigger PTI-like immune responses for defense [2,6]. That is to say, an efficient defense response requires specific recognition of the pathogen/herbivore and translation into defense signaling to regulate diverse cellular processes, including transcriptional rewiring and metabolic re-programming [7]. In plant immune defense networks, plants produce a huge array of different specialized metabolites, many of which are assumed to function as defenses. The classifications for specialized metabolites underwent multiple revisions during their history [8]. An extensive diversity of metabolites across the plant family have been conventionally classified in groups of ‘primary metabolites’, ‘secondary metabolites’, and ‘phytohormones’ [9]. The ‘secondary metabolites’ are also termed ‘specialized metabolites’ because they are often produced by specific plant lineages, including phenylpropanoids, terpenoids, alkaloids, sulfur-containing compounds, and modified fatty acids, which perform mostly protection functions that allow plants to adapt to their environment [10,11,12]. However, ever-increasing genetic and chemical studies have reported that these secondary metabolites could function as potent plant growth regulators [13]. So, a recently published review has suggested that the functional separation between the three groups is becoming less clear [7]. Thus, one of their important perspectives is that multifunctionality seems to be a widespread property of specialized metabolites.

Tea plants (*Camellia sinensis* (L.) O. Kuntze) belong to *Thea* of genus *Camellia* in the family Theaceae. In nature, tea plants are widely planted in southern China, Indian Assam, and southeast Asia, and they suffer from severe biotic stresses, such as herbivore and pathogen attacks, incurring severe losses of yield, and low-quality tea products [14,15,16]. In tea plants, hundreds of different bioactive metabolites are accumulated, of which the most enriched natural products are catechins, L-theanine, and purine alkaloids. In addition, flavonols, anthocyanins, and triterpenoid saponins also occur in large amounts. These compounds are the representative metabolites and play a major role in the tea’s flavor and health functions [17,18,19]. Among these compounds, catechins, caffeine, and volatile compounds have been found to play defensive roles against tea pests and diseases [20,21,22]. Tea geometrids are the main leaf-feeding pests in most tea plantations, including two sibling species, *Ectropis grisescens* Warren (Lepidopotera: Geometridae) and *Ectropis obliqua* Prout (Lepidopotera: Geometridae), which dramatically inflict harm to tea plantations [23] (Figure 1A). Another pest, the tea green leafhopper, *Empoasca onukii* Matsuda (Hemiptera: Cicadellidae), is the most disruptive pest across tea plantations. The nymphs and adults pierce tender tea shoots and suck the sap, and the adults oviposit in the tender tissues, seriously affecting tea production and quality [24,25] (Figure 1B). Among the diseases of tea plants, blister blight disease is caused by the obligate fungus *Exobasidium vexans* Massee, which only infects the succulent young leaves and stems. It is one of the serious diseases of tea plants that occurs in almost all tea-growing countries [26] (Figure 1C). Anthracnose is the most widespread disease that occurs commonly on the leaves of tea plants, resulting in damage to the leaves (Figure 1D). This disease is caused by several species of the genus *Colletotrichum*, while *Colletotrichum camelliae* has been recorded in countries like China and Sri Lanka [20]. Therefore, whether the abundant specialized metabolites of the tea plant have biological functions and play roles in defense against two pests and two diseases is meaningful and important.

In this review, we surveyed the existing literature for specialized metabolites elicited by herbivore infestation and pathogen infection and then summarized the defensive roles and action modes of specialized metabolites along with the intrinsic molecular mechanisms in tea plants. Further, we also pointed out important directions for future research. Therefore, we hope this review will help better understand tea plant herbivore/pathogen interactions and the function of integral metabolite regulatory networks. Moreover, the multifunctionality of metabolites can provide new explanations for tea plant defense under constant pressure from various pathogens and pests in their natural environment.

## 2. The Phytohormones Involved in Pest/Pathogen-Induced Defense in Tea Plants

Plants have evolved sophisticated defense mechanisms to protect themselves from biotic and abiotic stresses. In defense processes, the activated phytohormonal signaling networks connect plant perceptions and early signaling transduction to broad transcriptional reorganization and metabolite production, playing a central role in plant defense responses [27,28]. Many findings have suggested that plant defense responses are fine-tuned by phytohormones, including well-established jasmonic acid (JA), salicylic acid (SA), ethylene (ET), and other critical signals, such as gibberellin (GA), abscisic acid (ABA), brassinosteroid (BR), cytokinin (CK), auxin (indole-3-acetic acid, IAA), and so on [29,30,31,32,33]. Among them, JA is the core signaling pathway that regulates herbivore/pathogen-induced defense, functioning as a key player in regulating defensive metabolite production [34,35].

In tea plants, JA and SA are the principal signaling molecules that activate defense pathways, and other phytohormones are required to ensure the proper coordination of growth and defense. For example, multiple studies have consistently found that the JAs (JA, JA-Ile, and 12-oxo-10,15(Z)-phytodienoic acid (OPDA)), SA, IAA, and ABA were activated upon the infestation of *E. grisescens* and *E. onukii* (Table 1) [36,37,38,39]. Furthermore, a study found that JA levels significantly increased, whereas GA levels notably decreased in tea plants upon attack by *E. grisescens*, with an obvious antagonistic cross-talk between JA and GA signals [40]. The *E. onukii*-induced minor increase in JA level has been reported recently [41]. Additionally, exogenous application of JA enhanced the resistance of tea plants to *E. grisescens* by activating defense characteristics, including defense gene expression and the accumulation of defensive proteins and metabolites. For instance, pretreatment with MeJA increased PPO activity by activating the transcripts of *CsPPO2* and *CsPPO4* [40] and induced the biosynthesis of defensive catechins and volatiles [38,42]. In the tea diseases, *C. camelliae* infection significantly increased JA and OPDA content, and *E. vexans* infection increased the endogenous levels of JA and SA as well as the expression of synthesis-related genes [43,44] (Table 1). The levels of SA and SA glucose ester, along with the expression of the UDP-glycosyltransferase (UGT) gene *CsUGT87E7,* were significantly induced by gray blight infection, suggesting a positive role of the SA signal in tea resistance to pathogens [45]. 

In conclusion, the above studies suggest that (i) both JA and SA signaling pathways are the two key signals in tea plants. Similar to other plants, the attacks of chewing pests (*E. grisescens* and *E. obliqua*) and *C. camelliae* infection preferentially activate the JA pathway, and the infection of *E. vexans* significantly activates both JA and SA signals, whereas the attack of piercing-sucking *E. onukii* and the infection of gray blight preferentially activate the SA pathway [34,36,41,44]. Thus, an in-depth understanding of the molecular mechanisms underlying the herbivore/pathogen-induced phytohormone signaling in tea plants will require more extensive and comprehensive investigations in vitro and in vivo. (ii) Among these phytohormone signals, JA signaling is well established as the core pathway that regulates tea plant defense against herbivores and pathogens. The herbivore/pathogen-induced patterns of JA in tea plants, consistent with previous studies in other plant species, suggest that JA signaling is widely conserved among diverse plant species [27]. There have been numerous studies proving that the JA signaling pathway plays an essential role in protecting plants from multiple stresses, and JAZ-MYC modules have been studied precisely [48,49]. In tea plants, the JAZ-MYC modules have been verified to play a crucial role in response to low temperatures and *C. camelliae* infection [46,50], and intense research has revealed essential molecular components of the JA pathway [51,52]. The explicit modules and the underlying mechanisms still need elaborated investigations. In addition, few studies have been conducted on SA, ET, ABA, GA, and BR signaling pathways in defense over the past decade; therefore, more research is requisite in these themes.

## 3. Defensive Functions of Volatiles Elicited by Pest/Pathogen Attacks in Tea Plants

Plant volatiles have important roles in many aspects of plant interactions with the environment, and more than 1700 volatile compounds have been characterized in plants [53]. A large number of studies have proposed that both pest and pathogen attacks could elicit a substantial amount of plant volatiles, which mediated the behaviors of herbivores and their natural enemies, the growth rate of pathogens, and plant resistance via activating specific signaling pathways, as well as triggering the plant–plant communications that made the neighboring plants more resistant to the subsequently coming herbivores [22,54]. According to the biosynthesis pathway, volatile compounds in tea plants can be divided into fatty acid derivatives, benzenoids, and terpenes, whose precursors mainly originate from both the cytosolic mevalonate pathway and the plastidic methylerythritol phosphate pathway [55]. Although the composition and emission amount of herbivore-induced plant volatiles (HIPVs) are influenced by the herbivore species and the attack degree, many common compounds are induced by different herbivores. For example, once attacked by *E. grisescens* or *E. onukii*, the tea plants released more than 30 or 20 volatile compounds, respectively, including (*Z*)-3-hexenol, (*Z*)-3-hexenyl acetate, linalool, indole, β-ocimene, α-farnesene, and (*E*)-nerolidol, etc., and these compounds have been documented to serve three distinct ecological functions: direct defense against insects, attracting insect predators or parasitoids, and signaling within or between tea plants [22,37,54,56,57,58,59] (Table 2). 

Compared with other defensive metabolites, an important feature is that HIPVs can respond to insect stress more rapidly. For example, volatiles (e.g., (*Z*)-3-hexenal, (*E*)-2-hexenal, and (*Z*)-3-hexenyl acetate) can be released within minutes after infestation by pests [37,55], which suggests that HIPVs could perform as defense signals. Over the past few years, many studies have focused on the signaling mechanisms of specific HIPVs, such as (*Z*)-3-hexenol, (*E*)-nerolidol, (*E*)-4,8-dimethyl-1,3,7-nonatriene (DMNT), and indole, which have been verified to play an important role in the induction of JA-dependent resistance against pests [37,54,56,57,62]. (*Z*)-3-hexenol is one of the most intensively studied HIVPs and exhibits multifaceted defense-related functions, and the defense mechanisms have been demonstrated in tea plants. Firstly, field and laboratory experiments have confirmed that (*Z*)-3-hexenol effectively activated the defense against tea geometrids and made tea plants more attractive to tea geometrid parasitoids. Secondly, more thorough investigations proved that (*Z*)-3-hexenol triggered JA and ET signaling pathways [54,60,64]. Thirdly, the glycosylation of the (*Z*)-3-hexenol further enhanced insect resistance [22]. Finally, (*Z*)-3-hexenol was a signaling molecule absorbed by adjacent healthy plants, and it would be converted into three insect defensive compounds to enhance the tea plant’s resistance [58]. Accordingly, it could be used as a chemical elicitor to explore the biological strategies to control tea geometrids.

Glycosylation is a key mechanism that determines the chemical complexity of metabolites in plants [65]. Glycosylation reactions are catalyzed by glycosyltransferases that transfer an activated nucleotide sugar to acceptor aglycons to form glycosides as well as sugar esters [66]. In tea plants, CsUGT85A53-1 catalyzed (*Z*)-3-hexenol to produce (*Z*)-3-hexenyl glucoside [18]. Moreover, the linalool and linalool oxides and their glucosides were accumulated after *E. onukii* and *E. grisescens* infestations [67]. These findings were identical to the results of previous studies in tomato and kiwi fruits [68,69], which imply that the glycosides of volatiles in plants act not only as direct defensive compounds but also as a source of volatile storage that would emit immediately after herbivory attack and then attract natural enemies of herbivores.

## 4. Defense Responses of Flavonoids to Herbivore/Pathogen Attacks in Tea Plants

Flavonoids are an important class of secondary metabolites involved in multiple aspects of plant defense against pathogens, herbivores, and ultraviolet [70]. As a class of specialized metabolites, phytoalexins and phytoanticipins belong to flavonoids. Phytoalexins are synthesized de novo after pathogen infections, and phytoanticipins are either constitutively present or synthesized from preformed constituents [8]. For example, the phytoalexin sakuranetin is a biologically important compound due to its antimicrobial activity and is induced only after pathogen infection in rice plants [71]; quercetin acts as a phytoanticipin to limit the establishment of biotrophic pathogens, thus delaying or reducing their sporulation [72]. In tea plants, flavonoids contain flavonols, dihydroflavonols, catechins (flavan-3-ols), flavanones, anthocyanidins, etc. Catechins are especially abundant in tender buds and leaves, and they play important roles in quality, flavor, and health value, as well as protecting plants against herbivores and pathogens [17,55]. From the perspective of biosynthesis, tea flavonoids originate from diverse branches of the phenylpropanoid pathway, whose precursor is the shikimate pathway-derived phenylalanine. The shikimate pathway takes place in the plastid and provides many essential substances and precursors for the biosynthesis of large biomolecules. For example, the shikimate pathway-derived gallic acid and glucogallin are the essential substrates for the synthesis of polygalloylated glucoses and galloylated catechins [17]. Several studies have shown that herbivore infestation activates the biosynthesis of flavonoids, which act as inducible defensive compounds (Table 3). For example, the studies suggested that a large number of genes involved in the biosynthesis of flavonoids were activated, and the contents of flavonols, dihydroflavonols, flavan-3-ols, anthocyanidins, flavones, and flavonoid glucosides, such as myricetin, rutin, dihydroquercetin, and dihydromyricetin, were elevated, but some flavonoid precursors and derivatives were decreased in tea plants upon herbivore attack [73,74]. Further investigation found that *E. grisescens* infestation significantly increased the accumulation of quercetin glucosides produced from quercetin catalyzed by UGT89AC1, and an artificial diet supplemented with quercetin glucoside reduced the larval growth rate [75]. Moreover, the contents of tricetin, kaempferol 3-O-glucosylrutinoside and methyl 6-Ogalloyl-b-D-glucose, as well as the expression levels of key genes pertaining to flavonoids biosynthesis, were significantly up-regulated during *E. onukii* infestation [41]. Additionally, flavonoids have been found to be elicited by pathogen infections. For example, *E. vexans* infection elevated the accumulation of quercetin and kaempferol glucosides and kaempferol triglycosides but substantially reduced the accumulation of apigenin and myricetin glycosides [76]. In summary, flavonoids responded specifically to different insects and pathogens; nevertheless, targeted metabolites and the underlying precise mechanisms of defense against insects and pathogens need in-depth investigations, according to the current studies.

Catechins are the dominant flavonoids in tea plants, classified as ester or non-ester types. The major non-ester type catechins include catechin (C), epicatechin (EC), gallocatechin (GC), and epigallocatechin (EGC), and the major ester-type catechins include epicatechin-3-gallate (ECG) and epigallocatechin-3-gallate (EGCG) [17]. The concentration of ester-type catechins is much greater than that of non-ester type [77,78]. Catechins not only have multiple effects on human health with antimicrobial, antiviral, and antiaging activities [79], but also have important defensive functions against herbivores and pathogens (Table 3). For example, the contents of C, EC, GC, ECG, and GCG were increased, while the levels of EGCG declined in damaged leaves after an *E. obliqua* attack during 3–24 h [73]. Our study used mechanical wounding supplemented with the regurgitant of *E. grisescens* to simulate herbivore feeding to observe the change in catechins during 24–72 h after treatment [38]. The accumulation of C, EC, and EGCG was significantly augmented compared to the mechanical wounding, and an artificial diet supplemented with them reduced larval growth rates in a dose-dependent and time-dependent manner. Both studies showed that C and EC accumulated significantly, while EGCG decreased within 24 h in the former study and increased between 24 and 72 h in our study. Among several possible explanations, we consider the following two aspects: (i) the level of EGCG may be influenced by elicitation time and treatment type; (ii) the structure of EGCG is complex owing to the number of hydroxyl groups in the B-ring and presence of a galloyl moiety, and its accumulation may be influenced by several factors, such as precursors, degradation, or polymerization due to autoxidation. In addition, several reports have shown that the accumulation of catechins was elicited by *E. onukii* infestations and pathogen infections. During *C. fructicola* infection, the contents of C and EGCG were elevated, and in vitro, catechins inhibited mycelial growth in a dose-dependent manner [20]. Furthermore, *E. vexans* infection induced the accumulation of C, EC, EGC, and EGCG. These results suggested that herbivory and pathogen attacks elicited the differences in catechin metabolism [76]. Among seven catechins, both C and EGCG were effectively triggered upon attack by *E. grisescens*, *E. obliqua*, *E. onukii*, *C. fructicola*, and *E. vexans*, which suggested that C and EGCG could respond conservatively to different herbivores and pathogens, while other components responded differently. These findings align with many other studies, which have reported the diverse roles of EGCG in plant–environment interactions [80]; for instance, EGCG has antibacterial, antifungal, and anti-herbivore properties due to altering the metabolism of folic acid in bacteria and fungi or enhancing plant resistance against diverse diseases and herbivores [81,82]. Most studies have focused on the changes in the accumulation of catechins in response to stress; although only a few studies have tried to verify the defense functions, the underlying mechanism is seldom studied.

## 5. Defense Responses of Caffeine, Theanine, and Amino Acids to Pest/Pathogen Attack in Tea Plants

As the most well-known purine alkaloid, caffeine (1,3,7-trimethylxathine) accumulates at higher levels in the tea bud and young leaves, and its biosynthesis involves several critical methylation reactions catalyzed by N-methyl transferases and a 7-methylxanthine nucleosidase [83]. Caffeine is not only closely related to nitrogen metabolism, but it also plays an important role in the direct defense against insect herbivores and pathogens [84,85,86,87]. An *E. obliqua* attack activated the caffeine biosynthesis and increased its accumulation, while an *E. onukii* infestation did not change the levels of caffeine. A *C. fructicola* infection induced caffeine accumulation, and caffeine strongly inhibited mycelial growth by affecting the mycelial cell walls and plasma membranes in vitro; however, an *E. vexans* infection reduced caffeine levels [20,41,73,76] (Table 3). These studies suggest that caffeine responds differently to different insects and pathogens.

Theanine is a nonprotein amino acid with the highest content in tea plants, accounting for 1–2% of dry tea and more than 50% of total free amino acids [55]. Theanine is mainly distributed in roots, followed by young leaves, stems, flowers, and old leaves, and it is synthesized from L-glutamic acid and ethylamine by the catalytic action of theanine synthase [17]. Theanine is closely related to nitrogen assimilation and metabolism [86,88]. In light of the important association between basic amino acids and theanine and nitrogen metabolism, the levels of theanine and eight basic amino acids (glutamate, serine, cysteine, tyrosine, methionine, phenylalanine, glycine, and lysine) were variably induced upon the attack of *E. obliqua*. While the contents of predominant amino acids, such as theanine, glutamate, aspartate, serine, and glutathione, were down-regulated by the infestation of *E. onukii* [41,73]. These studies suggested that theanine, glutamate, and serine responded conversely to *E. obliqua* and *E. onukii*. In plants, a high carbon flux is committed to the biosynthesis of phenylalanine, tyrosine, and tryptophan, owing to their roles not only in the production of proteins but also as precursors to thousands of primary and specialized metabolites. Of the three amino acids, the major carbon flux proceeds toward phenylalanine; its derivatives include flavonoids, isoflavonoids, tannins, anthocyanins, and volatiles, and tyrosine serves as the precursor for quinones, betalains, and isoquinoline alkaloids [89,90]. *E. obliqua* infestation induced the accumulation of phenylalanine and tyrosine, suggesting that phenylalanine and tyrosine may be involved in the biosynthesis of specialized metabolites, such as flavonoids, volatiles and alkaloids, which have been proven in tea plants.

## 6. Conclusions and Perspectives

This review summarized the progress of recent research regarding the defensive function of specialized metabolites against herbivores and pathogens in tea plants. In these studies, some direct evidence of phytohormones, volatile compounds, and flavonoids serving as defense compounds against herbivores and pathogens has been obtained in vitro, and the defensive functions and action mode along with the intrinsic molecular mechanisms have been partly elucidated, long lagged behind those of model plants mainly due to the complicated genetic background and immature transformation system. In current research, one key question is which specialized metabolites are the potential key defense agents and what the regulation mechanism is. JA and SA are considered predominant hormones in defense against pests and pathogens, which regulate the biosynthesis of specialized metabolites such as benzyl nitrile and indole via JAZs-MYCs interactions. The other way around, volatiles, such as (*Z*)-3-hexenol, indole, I-nerolidol, and DMNT, can activate JA, ET, ABA, and other hormone signaling pathways and regulate defense gene expression and defense metabolite accumulation, then enhance plant resistance (Figure 2). Thus, as an important defense factor, the functions of volatiles are as follows: (i) direct anti-pest and anti-pathogen activities; (ii) enhancing plant resistance by activating hormone signaling; (iii) being converted into resistance-related glycosides. Moreover, EGCG has the same action model in defense as volatiles, and EGCG from galloylated EGC has anti-insect/pathogen activity and can regulate JA signal and methylester enzyme activity to enhance plant resistance to pests and disease. In our opinion, these key defensive metabolites not only have anti-insect/disease activity but also stimulate signaling pathways to activate plant defense and jointly mediate plant resistance to insect pests and pathogens. These metabolites play crucial roles in the process of tea plant defense against insect pests and have been assumed to act as key defenders, but the specific mechanisms for their synthesis and regulation pathways need to be further investigated.

Further research is needed to improve the following aspects: The glycosylation of volatile compounds is one of the important mechanisms in tea plants, but relatively little is known regarding the function of the glycosylation and its mechanism. In addition, the synthesis pathways of flavonoids have been well studied in tea plants, such as a highly conserved MYB-bHLH-WD40 (MBW) transcription complex that regulates flavonoids synthesis, but there are few reports on defense-related transcription complex components. Thus, with the continuous development of experimental techniques, modern high-throughput technology, and multi-omics analysis techniques, each aspect above will require further in vivo and in vitro evidence for further exploration of relevant mechanisms. In a word, we summarize many potential markers for tea plant resistance against pests and pathogens and improve our understanding of the defense mechanisms of plants.

## Figures and Tables

**Figure 1 plants-13-00323-f001:**
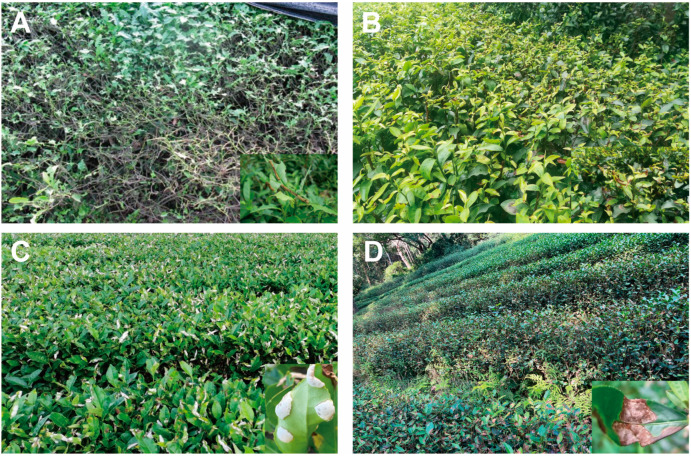
Symptoms of *Ectropis grisescens* infestation (**A**), *Empoasca onukii* infestation (**B**), *Exobasidium vexans* infection (**C**), and *Colletotrichum camelliae* infection (**D**) in the tea plantations.

**Figure 2 plants-13-00323-f002:**
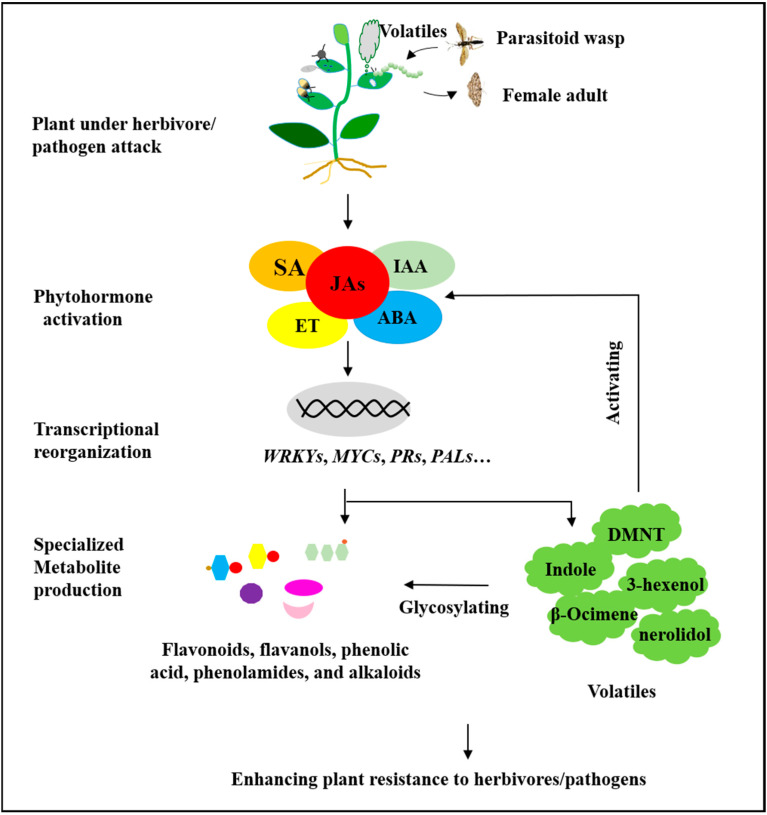
Schematic drawing of the mechanism underlying specialized metabolite-mediated resistance to pests and pathogens in the tea plant (*Camellia sinensis*).

**Table 1 plants-13-00323-t001:** Defense responses of phytohormones in tea plants to pest/pathogen attack.

Pests/Pathogens	Phytohormones	Defense Responses	References
*E. grisescen*/ *E. obliqua*	JA, SA, ET, IAA, ABA	All five phytohormone contents were elevated following herbivory, and exogenous treatment of JA, IAA, and ACC enhanced the resistance of tea plants to *E. grisescens*,	[38,39]
*E. onukii*	SA, JA	SA content increased significantly and JA content increased slightly following herbivory,	[41]
*C. camelliae*	JA, IAA	JA and IAA content significantly increased after infection, and exogenous application of MeJA modulates tea plant susceptibility to *C. camelliae.*	[43] [46]
*E. vexans*	JA, SA	JA and SA content significantly increased after infection.	[44,47]

*E. grisescen*: *Ectropis grisescens*; *E. obliqua*: *Ectropis obliqua*; *E. onukii*: *Empoasca onukii*; *C. camelliae*: *Colletotrichum camelliae*; *E. vexans*: *Exobasidium vexans*; ACC: 1-aminocyclopropane-1-carboxylic acid (the precursor of ethylene).

**Table 2 plants-13-00323-t002:** Ecological functions of volatiles in defense against pests/pathogens in tea plants.

Volatiles	Targets	Functions	References
(*Z*)-3-hexenol	*E. grisescens* *E. obliqua*	Enhanced direct and indirect tea resistance by activating JA and ET signaling, reducing the performance of herbivores, and making tea plants more attractive, the main parasitoid wasp, and the (*Z*)-3-hexenyl-glycoside had anti-insect activity via reducing the larval growth rate.	[54,58]
(*E*)-nerolidol	*E. onukii* *C. fructicola*	Enhanced tea resistance by activating JA and ABA signaling and increasing the accumulation of defensive compounds, thus reducing the performance of herbivores. In addition, inhibited hyphal growth.	[60]
Indole	*E. obliqua*	Primed tea resistance by JA signaling and defense-related secondary metabolites reduces the growth rate of herbivores.	[37]
β-Ocimene	*E. obliqua*	Enhanced tea resistance by activating the signal pathway and reducing the weight gain of herbivores; in addition, strongly repelled mated females in behavioral bioassays.	[61] [57]
Benzyl nitrile	*E. grisescens*	Inhibited larval growth in vitro and repelled larvae in behavioral bioassay.	[62]
Geraniol	*C. camelliae*	Inhibited the growth of *C. camelliae* by decreasing the activity of the defense enzymes.	[63]
DMNT	*E. obliqua*	Promotes the resistance of neighboring intact plants by activating JA signaling.	[56]

*E. grisescen*: *Ectropis grisescens*; *E. obliqua*: *Ectropis obliqua*; *E. onukii*: *Empoasca onukii*; *C. camelliae*: *Colletotrichum camelliae*; *C. fructicola*: *Colletotrichum fructicola*; *E. vexans*: *Exobasidium vexans*; DMNT: (E)-4,8-dimethyl-1,3,7-nonatriene.

**Table 3 plants-13-00323-t003:** Defense responses of flavonoids, caffeine, theanine, and amino acids in tea plants to pest/pathogen attacks.

Metabolites	Pests/Pathogens	Defense Responses	References
Flavonoids	*E. obliqua*	Significantly increased the contents of myricetin, rutin, dihydroquercetin, and dihydromyricetin.	[73]
	*E. grisescens*	Increased the accumulation of quercetin glucosides, and an artificial diet supplemented with quercetin glucoside reduced the larval growth rate (identified anti-herbivore function).	[75]
	*E. onukii*	Significantly upregulated the levels of tricetin, kaempferol 3-O-glucosylrutinoside, and methyl 6-Ogalloyl-b-D-glucose.	[41]
	*E. vexans*	Increased levels of quercetin, kaempferol glucosides, and kaempferol triglycosides, and decreased levels of apigenin and myricetin glycosides.	[76]
Catechins	*E. obliqua*	Increased the contents of C, EC, GC, ECG, and GCG, and declined the content of EGCG.	[73]
	*E. grisescens*	Significantly increased the contents of C, EC, and EGCG, and the artificial diet supplemented with C, EC, and EGCG reduced larval growth rate (identified anti-herbivore function).	[38]
	*E. onukii*	Induced the accumulation of EC, EGC, EGCG, ECG, EGC-ECG dimer, and EC-ECG dimer.	[41]
	*C. fructicola*	Induced the accumulation of C and EGCG, which strongly inhibited the growth of the mycelium (identified anti-pathogen function).	[20]
	*E. vexans*	Induced the accumulation of C, EC, EGC, and EGCG.	[76]
Caffeine	*E. obliqua*	Activated biosynthesis and accumulation of caffeine.	[73]
	*E. onukii*	Presented no change in response to the herbivore attack.	[41]
	*C. fructicola*	Induced caffeine accumulation inhibited mycelial growth by affecting mycelial cell walls and plasma membranes.	[20]
	*E. vexans*	Reduced the content of caffeine.	[76]
Theanine amino acids	*E. obliqua*	Increased the levels of theanine and eight basic amino acids (glutamate, serine, cysteine, tyrosine, methionine, phenylalanine, glycine, and lysine).	[73]
	*E. onukii*	Decreased the levels of theanine, glutamate, aspartate, serine, and glutathione.	[41]

*E. grisescens*: *Ectropis grisescens*; *E. obliqua*: *Ectropis obliqua*; *E. onukii*: *Empoasca onukii*; *C. fructicola*: *Colletotrichum fructicola*; *E. vexans*: *Exobasidium vexans*.

## Data Availability

This study did not report any data.

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
