# Peer review of "Recent Advances in the Specialized Metabolites Mediating Resistance to Insect Pests and Pathogens in Tea Plants (Camellia sinensis)"

_plants, 2024, doi:10.3390/plants13020323_

Round 1
Reviewer 1 Report
Comments and Suggestions for Authors
Dear Authors
i mentioned some comments in the attached file track revision

Comments on the Quality of English LanguageThe English needs some few corrections
Reviewer 2 Report
Comments and Suggestions for Authors
This review presents the role of specialized metabolites involved in resistance to pests and pathogens in tea plant as clearly indicated in the title. It is well organized and gathers a lot of information about tea plant. However, being a review on tea plant defense, the concept of innate immunity should be introduced since the signaling transduction pathways involve hormones and responses leading to plant defense. Also, the terms of systemic induction, ISR and SAR should be present in this review.
Additional suggestions of improvement are i) references to model plant mechanisms, that are basically the same in all plants; and ii) phytoanticipin/phytoalexin as indicated for flavonoids; iii) any references to PGPR as agents triggering systemic induction: this will add a nice overview on other ways to trigger tea secondary metabolism, increasing the scope and interest of the review.
Specific comments follow below.
P2, L50. Indicate reference for classification of metabolites: primary, secondary and phytohormones
Table 1. scientific binomials in bold
Table 2. Spell out DMNT
P5, L180, adjust punctuation. Full stop or comma?
L192. Include “of” the accumulation OF quercetin-glucosides
L178. Defense responses of flavonoids. This section will benefit from the concept phytoanticipins and phytoalexins, being phytoanticipins the metabolites that accumulate prior to triggering stress and phytoalexins the defensive compounds accumulating upon challenge. This could contribute to unveiling mechanisms of specific flavonoids in defense.
L273. Conclusion and discussion: this section should be kept to conclusions. There is no discussion in a review. Conclusions need to be short sentences and precise ideas.
L275. Capital letter
Reviewer 3 Report
Comments and Suggestions for Authors
The manuscript with the title “Recent advances in the specialized metabolites mediating resistance to pests and pathogens in tea plant (Camellia sinensis)”, is a comprehensive review on specialized metabolites from tea plantations as they are related to pests and pathogens. Due to the economic importance of this crop, this review could be a good reference work in the field. The authors relied on many recent sources. However, moderate improvements in English style, syntax and grammar could increase the readability.
Keywords: species name is misspelled “sinensi”, please correct
Also, note that keywords should not repeat words already mentioned in the title.
The Introduction provides a good background for the review.
However, I strongly suggest a paragraph somewhere in the introduction or at the discussion where it is clarified the state of knowledge on the main biosynthesis pathways (some are synthesized in the plastids some in cytosol), localization of the specialized metabolites (if trichomes and glandular hairs exist) or mechanism of release. I think this would connect very nicely all the information together. Some of these aspects are summarized in Conclusions and Discussion section, but are not entirely complete.
Particular notes:
Table 1 species name with italics.
Lines 116, 253 – in vitro/in vivo with italics
Line 132 – perhaps “implicated” could be better replaced with “proposed” or “inferred”?
Line 143 – which “above studies”? do you mean sources from table 2? Table 2 is below not above the line 143. Please clarify what you mean.
Line 203 – perhaps in-depth instead of in-deep
Table 1, 2, 3 – I am not sure that genus should be abbreviated in this case. Because the examples are of species from different genera. I suggest there should be explanatory footnotes for abbreviations or compounds too. Because tables should be fully self-explanatory.
References are predominantly new, which is good.
Best regards.
Comments on the Quality of English Languagemoderate improvements in English style, syntax and grammar could increase the readability.
